# Long-Term Outcomes of Crooked Nose Correction: The Value of Instrumental Diagnosis Trough Nasal Grid Analysis

**DOI:** 10.3390/diagnostics15091121

**Published:** 2025-04-28

**Authors:** Riccardo Nocini

**Affiliations:** Department of Head and Neck, ENT Department, School of Medicine, University of Verona, 37129 Verona, Italy; riccardo.nocini@univr.it

**Keywords:** crooked nose, deviated nose, rhinoplasty, nasal grid

## Abstract

**Background**: Rhinoplasty is a leading cosmetic surgery, with the crooked nose being one of the most complex challenges due to its combination of functional and aesthetic issues. Involving multiple nasal components, a crooked nose remains difficult to correct, with a high recurrence rate. The aim of this study was to analyze the long-term outcomes and stability of the nasal pyramid after surgery through a nasal grid analysis. **Methods**: A retrospective study was conducted on 360 patients (20 men and 16 women) treated for moderate to severe nasal deviation between January 2014 and January 2020. All patients underwent surgery by the same expert surgeon, with follow-ups extending to at least 24 months. Medical records, physical exams, and standardized photographic evaluations were used to assess outcomes. A nasal grid was adapted to analyze the long-term stability of nasal corrections, focusing on individual post-operative changes. **Results**: The study involved 360 patients, mostly men aged 22 to 65, with the majority having nasal deviations caused by extrinsic trauma (e.g., sports injuries). Nasal deviations affected various parts of the nose, and all patients underwent follow-up evaluations using a nasal grid to compare pre- and post-operative measurements. The results showed long-term stability in nasal corrections, with minimal changes observed between 1 month and 24 months post-surgery. Only one case had partial recurrence, requiring revision surgery, which was successful. **Conclusions**: The surgical correction of a crooked nose is complex and requires a personalized approach, particularly for severe septal deviations. Complete anatomical reconstruction, using advanced techniques like extracorporeal septal reconstruction and laser osteotomies, provides stable long-term results. The adapted nasal grid proved to be not only an effective post-operative assessment tool but also shows potential for the pre-operative classification of nasal deformity complexity. Although this study focused on a standardized surgical technique, future comparative analyses with alternative approaches are recommended to further validate the outcomes.

## 1. Introduction

Rhinoplasty is one of the leading cosmetic surgeries performed by plastic surgeons and the crooked nose is the most complex and challenging deformity of the nose to treat due to the simultaneous involvement of functional and aesthetic elements. Crooked nose is a generic term used to define all deformities which involve the nasal pyramid deviation in relation to the facial medio-sagittal plane [1]. This condition not only affects the physical symmetry of the face but also contributes to compromised nasal airflow, impacting the quality of life of affected individuals. It is important to recognize that a crooked nose is often a deformity involving multiple components [2]. These include a bony nail pyramid, upper and lower lateral cartilages and, especially, the nasal septum [3,4], (Figure 1).

Depending on the individual patient’s anatomy, the deviation may be related to bony pyramid pathology or septal deformity; most patients present with a combination of the two pathologies [5,6]. In most cases, the main component of the crooked nose is the extremely deviated nasal septum [7,8]. In these cases, it can appear vaguely C-shaped or S-shaped or wholly displaced to one side or the other. The etiology of crooked noses is diverse, ranging from congenital anomalies to acquired injuries resulting from trauma, such as sports accidents or vehicular collisions. Congenital cases and irregularities in cartilage growth centers during development can result in asymmetry. Acquired cases, on the other hand, frequently involve fractures or dislocations of nasal bones and cartilages, often producing irregular healing patterns that exacerbate the deviation. Additionally, previous surgical interventions can sometimes contribute to residual or secondary deformities, complicating future correction attempts.

Correcting a crooked nose remains a challenge, especially being a type of deformity with a high degree of recurrence. Addressing the complexities of a crooked nose requires a nuanced approach that takes into account both aesthetic and functional considerations [2,9]. Several factors contribute to the challenges inherent in surgical correction:Multifactorial Anatomy: The involvement of multiple structural components—including bone, cartilage, and soft tissue—necessitates a holistic approach to restoration.Intrinsic and Extrinsic Forces: Cartilaginous memory, the natural recoil tendency of cartilage to its original position, presents a significant hurdle. Additionally, external forces, such as scar contraction and soft tissue tension, may counteract surgical corrections.Recurrence Risk: Despite advanced techniques, crooked noses remain prone to post-operative recurrence, particularly in cases involving severe deformities or poor tissue quality.

There are several schools of thought in the treatment of crooked nose: (1) camouflage techniques [10,11,12]; (2) complete deconstruction and anatomic reconstruction of the nose [8,13]; and (3) a combination of techniques [14]. The most used and preferred technique we use is the complete anatomical reconstruction of the nose and, if necessary, the camouflage techniques.

Since it is a corrective surgical procedure subject to a higher frequency of recurrence than other surgical interventions, and since the literature is lacking regarding the topic of this paper, we wanted to examine the long-term outcomes of the corrections of crooked noses performed in our University Hospital G.B. Rossi in Verona (Italy), examining the long-term stability of the surgical corrections.

To conduct a meticulous examination, we revised and adapted a nasal grid, designed to standardize injection procedures in non-surgical rhinoplasty [15,16,17], in order to have precise anatomical nasal landmarks and to report the results of the clinical cases included in the study.

## 2. Materials and Methods

### 2.1. Patients

A retrospective study was carried out on 360 patients, 210 men and 150 women, treated between January 2014 and January 2020. The mean age of the patients was 40 (range: 22–65). Inclusion criteria were a moderate to severe degree of nasal deviation. The study included only patients with at least 24 months of follow-up.

Exclusion criteria were represented by metabolic, cardiologic, dermatologic, and oncologic illnesses as well as patients taking on a regular base corticosteroid, immunomodulators, and other medications that could interfere with the surgical process and cartilage biology.

All patients underwent the surgical intervention by the same expert surgeon between January 2014 and January 2020. For each patient, written informed consent was obtained, covering the processing of personal data and ensuring privacy protection.

### 2.2. Analysis

During the initial consultation, the patients’ medical records were reviewed for their ages, sexes, detailed medical histories (e.g., trauma, previous surgeries, and breathing difficulties), and the type of deformities they had. Physical and nasal endoscopic evaluations were used to identify deviations and irregularities of the bony and cartilaginous structures of the nose, as well as potential pathologies of the nasal turbinates. High quality photographic evaluation was performed at six moments using a Canon EOS 5D Mark IV, Canon EF 100 mm f/2.8 Macro USM (Canon Inc., Tokyo, Japan): pre-operative, 1 month, 3 months, 6 months, 1 year, and 2 years. The method of acquiring photographs was standardized using precise reference points within a photographic studio to place patients in the same position at each clinical check-up.

### 2.3. Nasal Grid

To perform a more precise analysis of the long-term post-operative stability of the correction of the crooked nose and its changes over time, we have revised and adapted the nasal grid [15] we designed for nasal injection procedures, in order to have standardized anatomical nasal landmarks (Table 1, Figure 2). The goal of this grid was to analyze the post-operative stability of the nose in the frontal view in order to detect eventual deviation recurrence. In Table 2 and Table 3, an explanation of the different lines used to build the grid is provided. Of course, nasal grid analysis is individual to each patient and cannot be used to do a comparison between different patients.

### 2.4. Surgical Technique (Figure 3A–J)

Surgery was performed via an external rhinoplasty approach. When we correct the deviated nose, the general surgical principles suggested by previous researchers are followed. The surgical sequence employed is initiated with an infiltration of L-bupivacaine 8.5% 10 mL mixed to carbocaine 2% 10 mL and 1 mL of adrenaline. We used between 6 to 10 mL of local anesthetic and waited for 20 min. First, we performed correction of the nasal septum trough the complete release of all the nasal structures from it. The bony pyramid was then corrected through several osteotomies performed with Er-Yag laser [18] (medial, lateral, and percutaneous nasal radix osteotomy). Frontal asymmetries of the dorsal septum were corrected with grafts (spreader grafts, alar batten, and septal bone grafts). In some patients with type III and V deviations in which a straight septal tilt was present, we severed the dorsal strip of the L-strut and overlapped the proximal and distal segments, and then we fixed them together using 5-0 polydioxanone sutures. In cases with severe deviations of the septal cartilage, involving both dorsal and caudal portions of the L-strut, extracorporeal reconstruction of the septal cartilage was used. When there was convexity in the caudal septum, a swinging-door maneuver and fixation suture or septal batten graft were used.

**Figure 3 diagnostics-15-01121-f003:**
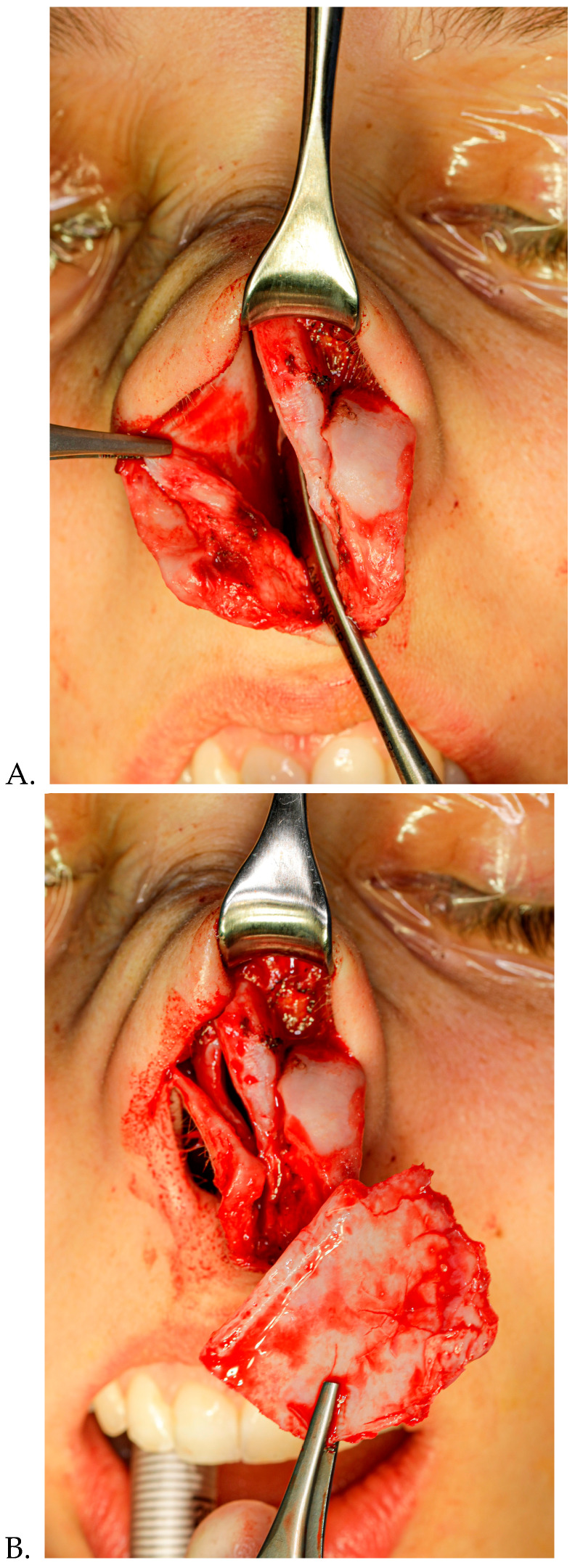
(**A**–**J**) Surgical phases of the correction of a crooked nose.

## 3. Results

A total of 360 patients were included in the study during the 5-year period. All the patients belonged to the age group between 22 to 65 years with a majority of the cases being male. The majority of the cases were due to extrinsic trauma (sports injuries) and fall injuries (45% as a result of sports injuries, 25% via home trauma, and 15% via traffic accidents). Only 1 of the 360 cases did not give any history of extrinsic trauma (congenital). All cases presented with external deviation of the nose, involving either the bony third (8 cases), the mid cartilaginous portion (5 cases) or both the mid and lower third (2 cases) or the bony and mid third (3 cases) along with nasal obstruction. All patients performed follow-ups correctly until the last post-operative visit.

Photographs obtained at the pre-operative time, at 1 month and at 2 years were analyzed using the nasal grid with a comparison of the different parameters in all patients. An analysis of all the measurements obtained for each single parameter of the nasal pyramid position was then performed and the percentage variations of the same over the follow-up times were calculated. In Table 4 and Table 5, the measurements obtained using the nasal grid during the analysis of the case illustrated in Figure 2, with a comparison of the pre-operative and post-operative measurements. Subsequently, the mean of all measurements performed on all patients was calculated and reported as a percentage variation in Table 6 and Table 7.

The results showed that after the correction of the deviated nose, the position of the nasal pyramid with adequate technique is well preserved in the long term. On average, we observed small changes that were not statistically significant in the various measurements taken at 1 month and 24 months after surgery. This is due to the healing processes that occur following the surgery, the changes of which are observed especially in the first month, Figure 4.

In analyzing the nasal grid data across 360 patients, mean positional changes between 1 and 24 months post-surgery remained minimal (average variation < 3%), and none reached statistical significance. Due to the near-universal surgical success rate, statistical correlation between the use of grafts and recurrence could not be calculated. The only recurrence observed occurred in a congenital case, later corrected by revision surgery.

Only in one case of a 33-year-old boy was there a partial recurrence of the deviation of the nasal pyramid. The deviation was moderate-severe and was due to a congenital cause. The patient then underwent revision rhinoplasty surgery 24 months later with a successful outcome.

### Statistical Considerations

Given the low recurrence rate (0.28%), inferential statistics on surgical failure were not feasible. However, we conducted exploratory analysis to quantify positional changes between the 1-month (B) and 24-month (C) follow-ups using nasal grid parameters. Descriptive statistics, including mean differences, standard deviations, and percentage changes, were calculated. Additionally, we estimated Cohen’s *d* effect sizes to evaluate the clinical relevance of changes. Most parameters showed minimal mean differences (range: +0.1 mm to +0.4 mm) and low effect sizes (Cohen’s *d* < 0.3), indicating high stability. No parameter showed a statistically or clinically significant deviation from its 1-month post-operative position. The greatest relative change occurred in the “Tip to M” parameter, likely inflated due to its near-zero initial value. These results support the long-term positional maintenance of the nasal pyramid following anatomical reconstruction.

## 4. Discussion

The correction of a crooked nose remains a persistent surgical challenge due to the interplay of structural, functional, and aesthetic complexities. The surgical correction of nasal deviation, particularly when the nasal septum is severely affected, requires a precise and individualized approach. The nasal deviation itself can be produced by both extrinsic and intrinsic forces [13]. These natural forces present in the cartilage structures and soft tissues (shortened muscles and connective tissue on the crooked side) continue to act on the nose undergoing rhinoplasty and make it difficult to achieve an excellent result in the post-operative period. Another factor that can cause the nose to return to its crooked shape is the incomplete correction of the deviated nasal septum.

One of the most common residual defects in rhinoplasties are various degree secondary deviations of the nasal pyramid with consequent esthetic and functional defects. This makes the surgical rhinoplasty a challenging procedure, particularly in crooked nose defect corrections, which itself brings multiple defects in terms of symmetry and functional altered areas. This study highlights several advancements in the surgical correction of nasal deviations and that the complete anatomical reconstruction of the nose remains the preferred technique for correcting crooked noses. This approach involves detailed planning and execution to address deviations in the nasal septum, bony pyramid, and cartilaginous structures. The precision offered by Er-YAG lasers enhances the accuracy of bony realignments, reducing tissue trauma and post-operative swelling. This innovation has set new standards for achieving symmetrical nasal pyramids.

The findings suggest that when these methods are applied correctly, they offer substantial long-term stability. The data shows only minor changes in nasal pyramid position from 1 month to 24 months post-surgery, indicating that the corrections are generally maintained over time.

However, this study also underscores the difficulty of achieving permanent correction for cases with congenital deviations or severe pathology. The case of a 33-year-old male who experienced partial recurrence of nasal deviation highlights the ongoing challenges faced by surgeons. This case reinforces the importance of continuous monitoring and, when necessary, revision procedures to address residual or recurring deviations.

The adaptation of the nasal grid for measuring post-operative stability represents a significant advancement in evaluating surgical outcomes. The nasal grid allows for precise and standardized anatomical landmarks, facilitating accurate longitudinal assessments both in surgical planning and follow-up. The use of this tool to compare pre-operative and post-operative measurements provides a clear view of the surgical results and any subtle changes over time. Although the study’s results show overall stability, the detailed measurements and variations offer valuable insights into the nuances of surgical correction and the potential for slight deviations that might not be immediately apparent.

This study confirms that when complete anatomical reconstruction is performed using advanced techniques—including extracorporeal septal reconstruction and laser osteotomies—the long-term outcome is highly stable. The use of a standardized nasal grid provides a novel and practical method for tracking subtle anatomical changes post-surgery. Importantly, while the nasal grid was used retrospectively here, it holds strong potential as a pre-operative classification tool. Future prospective studies could adapt it to predict complexity, guide surgical planning, and stratify patient risk.

A further avenue for research lies in comparing the outcomes of this approach with alternative techniques. Although this study did not include such a comparison, the results establish a baseline against which other approaches might be evaluated. Also, while grafts (e.g., spreader, batten, and septal bone) were employed according to individual needs, the low recurrence rate did not allow for meaningful correlation analysis. Nonetheless, we highlight this as a key topic for multicentre trials and broader datasets.

While this study provides robust data on long-term outcomes, it also highlights some limitations. One limitation is the variability in individual healing processes and the natural forces that continue to act on the nasal structures post-surgery. The relatively small cohort of 360 patients limits the generalizability of these findings. Expanding the study population would provide more robust statistical validation. Congenital deformities and severe traumatic deviations remain challenging due to their intricate anatomical distortions.

Other limitations include the absence of a control group using alternative techniques, the reliance on a single-surgeon series, and the lack of statistical correlation between graft use and recurrence. Although measurement consistency was ensured through methodological rigor, formal inter-rater reliability testing is pending and should be pursued in future validations of the nasal grid method.

The persistence of such forces can contribute to the challenge of maintaining the corrected shape, especially in cases with severe initial deformities.

## 5. Conclusions

The surgical correction of a crooked nose remains a complex and challenging procedure, requiring a highly individualized approach, especially when addressing severe deviations of the nasal septum. This study demonstrates that long-term correction of crooked nose deformities using complete anatomical reconstruction yields stable outcomes when supported by structured evaluation tools like a nasal grid. The grid’s application not only enhances post-operative assessment but may also serve as a valuable pre-operative classification system. Although recurrence was rare, complex congenital cases may still benefit from individualized monitoring and potential revision. The integration of quantitative tools, combined with further comparative studies, will likely refine surgical strategies and improve patient outcomes in the future.

## Figures and Tables

**Figure 1 diagnostics-15-01121-f001:**
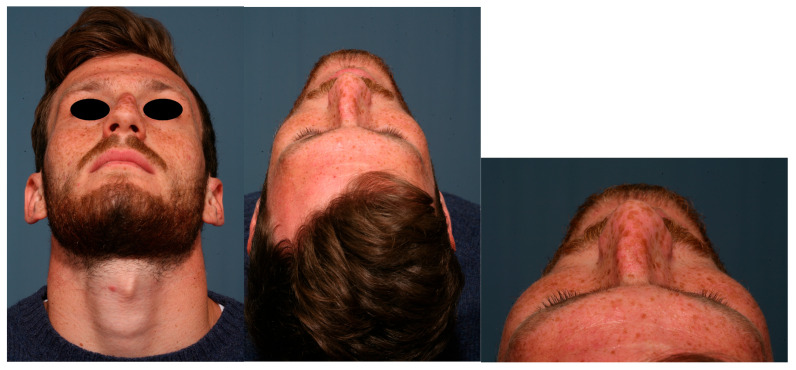
Crooked nose deformity (man, 21 years old).

**Figure 2 diagnostics-15-01121-f002:**
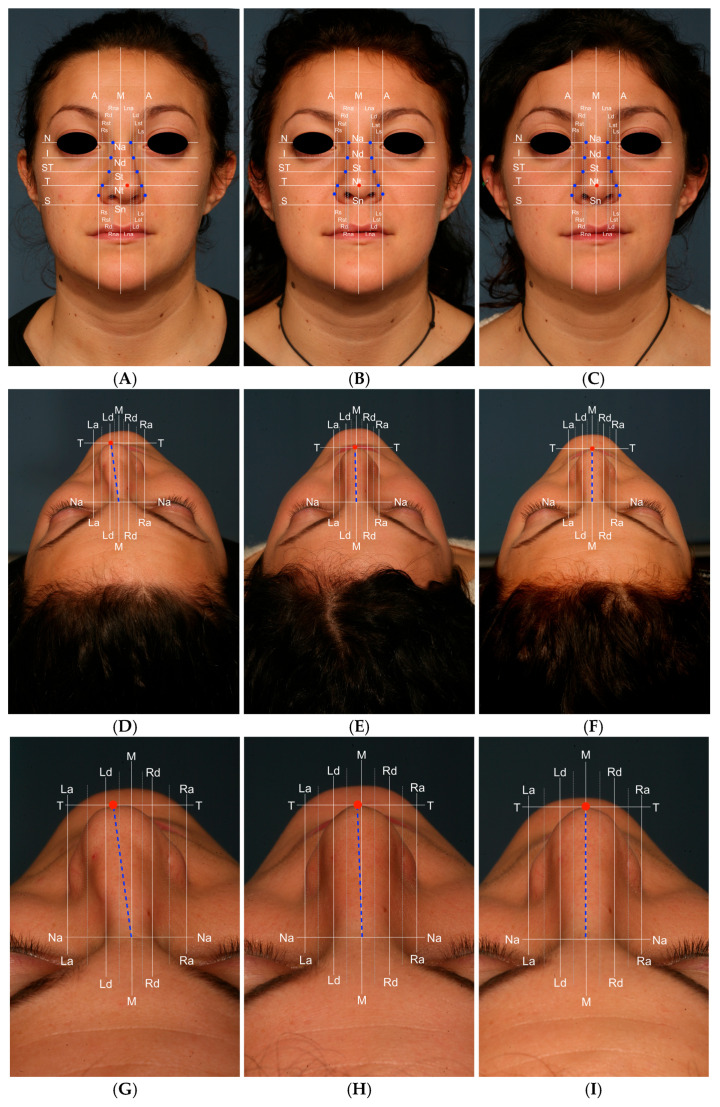
Explaining of the use of Nasal Grid to examine the anatomical landmarks of the nose in a standardized way. Crooked nose of a 25-year-old woman caused by a facial trauma: pre-op (**A**,**D**,**G**); 1 months after (**B**,**E**,**H**); 24 months after (**C**,**F**,**I**).

**Figure 4 diagnostics-15-01121-f004:**
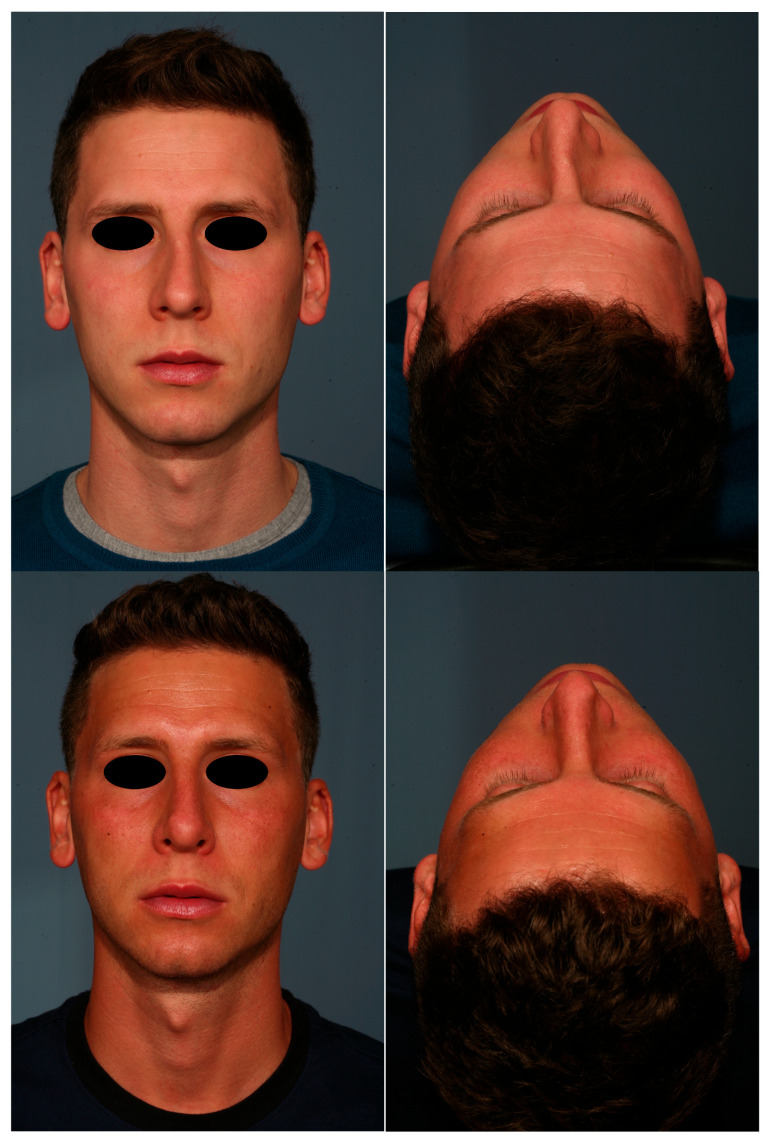
A 24 month follow-up in a patient of 27 years old with a deviation of the middle dorsum.

**Table 1 diagnostics-15-01121-t001:** Nasal landmarks in frontal illustration.

Landmark	Definition
Na	Nasion
Nd	Nasal dorsum
St	Supratip
Nt	Nasal tip
Sn	Subnasal
Rn	Right nasion
Ln	Left nasion
Rd	Right dorsum
Ld	Left dorsum
Rst	Right supratip
Lst	Left supratip
Rnt	Right nasal tip
Lnt	Left nasal tip

**Table 2 diagnostics-15-01121-t002:** Lines traced onto frontal illustration.

Line	Definition
**Vertical Lines**	
M (midline)	Through point M (midline interpupillary) to point Sn (subnasal point)
A (nasal ala)	Bilateral running through nasal ala insertion, parallel to M
**Horizontal Lines**	
N	Through nasion, perpendicular to line M
I	Through intermediate between N and ST, perpendicular to line M
ST	Through supratip, perpendicular to line M
T	Through nasal tip, perpendicular to line M
S	Through subnasale, perpendicular to line M

**Table 3 diagnostics-15-01121-t003:** Lines traced onto bird’s eye illustration.

Line	Definition
**Vertical Lines**	
M (Midline)	Through point M (midline interpupillary) to point Sn (subnasal point)
DR (Right Dorsum)	Through right side of nasal dorsum
DI (Left Dorsum)	Through left side of nasal dorsum
AR (Right Ala)	Through right nasal ala insertion
AI (Left Ala)	Through left nasal ala insertion
**Horizontal Lines**	
T	Through tip point, perpendicular to line M
Na	Through nasion point, perpendicular to line M.

**Table 4 diagnostics-15-01121-t004:** Results of the measurements of parameters in frontal view of the case illustrated in Figure 2 (A: pre-op, B: 1 months after, C: 24 months after).

Front (mm)			
	A	B (Difference Between A and B)	C (Difference Between B and C)
Rna	6.3	7.9 (+1.6)	8.0 (+0.1)
Lna	7.7	8.6 (+0.9)	8.8 (+0.2)
Rna	7.7	8.9 (+1.2)	9.0 (+0.1)
Lnd	10.1	10.1 (+0.0)	10.1 (+0.0)
Rst	9.2	10.3 (+1.1)	10.5 (+0.2)
Lst	12.6	11.8 (−0.8)	11.9 (+0.1)
Rnt	14.5	16.6 (+2.1)	16.7 (+0.1)
Lnt	16.5	15.2 (−1.3)	15.6 (+0.4)
Rsn	17.3	19.0 (+1.7)	19.0 (+0.0)
Lsn	18.8	18.9 (+0.1)	18.5 (−0.4)
Tip to M	5.2	0.3 (−4.9)	0.6 (+0.3)
Tip to A	6.3	18.5 (+12.2)	18.2 (−0.3)

**Table 5 diagnostics-15-01121-t005:** Results of the measurements of parameters in the focus of the nose of the case illustrated in Figure 2 (A: pre-op, B: 1 months after, C: 24 months after).

Focus Nose (mm)			
	A	B	C
MRa MLa	18.6	18.8 (+0.2)	18.9 (+0.1)
	21.8	19.2 (−2.6)	19.4 (+0.2)
MRd	6.9	9.4 (−2.5)	9.5 (+0.1)
MLd	8.8	8.76 (−0.04)	8.8 (+0.04)
T	6.3	1.9 (−4.4)	1.6 (−0.3)

**Table 6 diagnostics-15-01121-t006:** Average of measurements of nasal parameters in frontal view from 1 months after (B) to 24 months after (C).

Variation %	
	From B to C
Rna	+1.3%
Lna	+2.3%
Rna	+1.1%
Lnd	+0%
Rst	+1.9%
Lst	+0.8%
Rnt	+0.6%
Lnt	+2.6%
Rsn	+0%
Lsn	−2.1%

**Table 7 diagnostics-15-01121-t007:** Average of measurements of nasal parameters in Birdseye view from 1 months after (B) to 24 months after (C).

Variation Focus Nose %	
	From B to C
MRa	+0.5%
MLa	+1%
MRd	+1%
MLd	+0.4%

## Data Availability

The raw data supporting the conclusions of this article will be made available by the authors on request.

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
