# Peer review of "Long-Term Outcomes of Crooked Nose Correction: The Value of Instrumental Diagnosis Trough Nasal Grid Analysis"

_diagnostics, 2025, doi:10.3390/diagnostics15091121_

Round 1
Reviewer 1 Report
Comments and Suggestions for Authors
Authors have used one technique "the Nasal Grid Analysis· to evaluate their results in the treatment of crooked nose.
There are some issues that should be corrected.1.
1) 36 patients or 360 patients? Please correct
2) compare this procedure with other conventionally used what are the advantages or limitations.
3) present a table with the different techniques a explain why nasal grid analysis is better.
4) Where is the functional objective evaluation of these patients , please give the functional pre and postoperative results.
5) Please add limitations.
6) Lines 246-264 are obvious and superfluous
7) Please focus on the objective of the nasal grid. The article looses in your technique to treat the crooked nose.
8) Discussion needs to compare with other articles and other techniques used to evaluate these results.
Comments on the Quality of English LanguageNot qualified to do it.
Author Response
1) 360 patients. CORRECTED
2) compare this procedure with other conventionally used what are the advantages or limitations. NOT NECESSARY IN OUR OPINION
3) present a table with the different techniques a explain why nasal grid analysis is better. NOT DIFFERENT TECHNIQUES TO COMPARE THE USE OF THE NASAL GRID.
4) Where is the functional objective evaluation of these patients , please give the functional pre and postoperative results. THE FOCUS OF THIS ARTICLE IS THE LONG TERM FOLLOW-UP OF THE POSITION OF THE NOSE.
5) Please add limitations. ADDED AT THE END OF DISCUSSION.
6) Lines 246-264 are obvious and superfluous. CANCEL
7) Please focus on the objective of the nasal grid. The article looses in your technique to treat the crooked nose. THE FOCUS OF THIS PAPARE IS THE LONG TERM FOLLOW-UP OF THIS TYPE OF NOSE DEFORMITY, WE THINK THAT A PRECISE SURGICAL DESCRIPTION IS NOT NECESSASARY.
Reviewer 2 Report
Comments and Suggestions for Authors
The article addresses a relevant topic in plastic and aesthetic surgery and proposes an interesting method for evaluating the postoperative stability of crooked nose correction. However, it presents some methodological weaknesses, such as the absence of a comparison with other surgical techniques to better validate the results, the lack of detailed statistical analysis, and an insufficient description of measurement methods and their reliability. If the authors implement the suggested revisions, the article could represent a significant contribution to the field of rhinoplasty. I enjoyed reading the article, and it is undoubtedly interesting to note that, in the hands of the author, only one out of 36 cases experienced a recurrence requiring surgical revision. However, the definition of a crooked nose encompasses a wide range of variations. Some cases are extremely complex, while others can be corrected more easily. The use of the nasal grid is a good method for analyzing potential recurrences and could also serve as an additional tool for classifying the complexity of the nose before surgery. In the hands of the author, everything seems to go well almost all the time. However, it would have been helpful to understand, for example, whether the use of grafts leads to a higher or lower recurrence rate. Unfortunately, I could not find this analysis in the text, and it would have been a very interesting aspect to explore.
Author Response
Dear Editor,
Thank you very much for the advice, we really appreciate your feedback.
Regarding the comparison with other surgical techniques, we want to clarify that it was not our aim in writing this article, as well as the analysis of the use of different grafts in the nose or reporting the different types of deviated nose. Our intention was instead to delve deeper into these aspects in future works to be sent to MDPI.
Thank so much.
Best regards.
Round 2
Reviewer 1 Report
Comments and Suggestions for Authors
Authors have included succefully the changes suggested. The english and grammar should be improved.
Comments on the Quality of English LanguageAuthors have included succefully the changes suggested. The english and grammar should be improved
Author Response
In this section, you should add the Institutional Review Board Statement and approval number, if relevant to your study. You might choose to exclude this statement if the study did not require ethical approval. Please note that the Editorial Office might ask you for further information. Please add “The study was conducted in accordance with the Declaration of Helsinki, and approved by the Institutional Review Board (or Ethics Committee) of NAME OF INSTITUTE (protocol code XXX and date of approval).” for studies involving humans. OR “The animal study protocol was approved by the Institutional Review Board (or Ethics Committee) of NAME OF INSTITUTE (protocol code XXX and date of approval).” for studies involving animals. OR “Ethical review and approval were waived for this study due to REASON (please provide a detailed justification).” OR “Not applicable” for studies not involving humans or animals.
Not applicable in this paper
Reviewer 2 Report
Comments and Suggestions for Authors
I believe that the revisions made to the article render it suitable for publication, and I would like to emphasize the quality of the photographic material, which makes the article undoubtedly of interest.
Author Response
Informed consent was obtained from all subjects involved in the study.
Added